# Effects of Brown Seaweed (*Ascophyllum nodosum*) Supplementation on Enteric Methane Emissions, Metabolic Status and Milk Composition in Peak-Lactating Holstein Cows

**DOI:** 10.3390/ani14111520

**Published:** 2024-05-21

**Authors:** Dušan Bošnjaković, Sreten Nedić, Sveta Arsić, Radiša Prodanović, Ivan Vujanac, Ljubomir Jovanović, Milica Stojković, Ivan B. Jovanović, Ivana Djuricic, Danijela Kirovski

**Affiliations:** 1Department of Physiology and Biochemistry, Faculty of Veterinary Medicine, University of Belgrade, Bulevar Oslobodjenja 18, 11000 Belgrade, Serbia; dusan.bosnjakovic@vet.bg.ac.rs (D.B.); ljubomir.jovanovic@vet.bg.ac.rs (L.J.); stojicm@vet.bg.ac.rs (M.S.); ix@vet.bg.ac.rs (I.B.J.); 2Department of Ruminant and Swine Diseases, Faculty of Veterinary Medicine, University of Belgrade, Bulevar Oslobodjenja 18, 11000 Belgrade, Serbia; sreten.nedic@vet.bg.ac.rs (S.N.); sarsic@vet.bg.ac.rs (S.A.); prodanovic@vet.bg.ac.rs (R.P.); vujanac@vet.bg.ac.rs (I.V.); 3Department of Bromatology, Faculty of Pharmacy, University of Belgrade, Vojvode Stepe 450, 11221 Belgrade, Serbia; ivana.djuricic@pharmacy.bg.ac.rs

**Keywords:** dairy cows, enteric methane emissions, brown seaweeds, metabolism

## Abstract

**Simple Summary:**

Humanity faced the hottest year in observational history in 2023. If the trend of rising temperatures on Earth continues, irreversible changes in the environment will occur with negative consequences for the lives of us all. In order to limit the further increase in temperatures on Earth, it is necessary to reduce the emissions of greenhouse gases (GHGs) in various sectors. Since ruminant farming is a significant contributor to anthropogenic GHG emissions due to methane emissions, we attempted to reduce methane emissions in dairy cows by using brown seaweed (*Ascophyllum nodosum*) in cow rations. Indeed, we achieved a reduction in methane emissions, without disturbing the productivity of cows. We also found a more favorable metabolic status and milk composition in supplemented cows. Additional studies are needed to define precisely how brown seaweeds can be included into strategies to reduce methane emissions from the dairy industry.

**Abstract:**

The dairy industry contributes significantly to anthropogenic methane emissions, which have an impact on global warming. This study aimed to investigate the effects of a dietary inclusion of brown seaweed *Ascophyllum nodosum* on enteric methane emissions (EMEs), hematological and blood biochemical profiles, and milk composition in dairy cows. Eighteen Holstein cows were divided into three groups: CON (non-supplemented cows), BS_50_ (50 mL of 10% *A. nodosum*), and BS_100_ (100 mL of 10% *A. nodosum*). In each cow, measurements of EME, dry matter intake (DMI), and milk yield (MY), as well as blood and milk sampling with respective analyzes, were performed before supplementation (P1), after 15 (P2) days, and after 30 (P3) days of supplementation. *A. nodosum* reduced (*p* < 0.05) methane production, methane yield, and methane intensity in both BS_50_ and BS_100_, and raised DMI (*p* < 0.05) only in BS_50_. Total bilirubin (*p* < 0.05) was higher in BS_50_ compared to CON cows in P2, and triacylglycerols were lower (*p* < 0.05) in BS_50_ than in CON cows in P3. Higher milk fat content was found in BS_50_ than in CON cows in P3. C16:0 proportions were higher (*p* < 0.05) in BS_50_ and BS_100_ than in CON cows, while C18:3n-3 was higher (*p* < 0.05) in BS_100_ than in BS_50_ and CON cows in P3. Dietary treatment with *A. nodosum* reduced EMEs and showed the potential to increase DMI and to improve energy status as well as milk composition in peak-lactating dairy cows.

## 1. Introduction

The beginning of the 21st century was marked by growing concern about the anthropogenic contribution to global warming and climate change due to increased emissions of greenhouse gases (GHGs) into the Earth’s atmosphere [1,2]. Warning reports from the latest United Nations Climate Change Conference [3] and the World Meteorological Organization [4] indicated that global temperatures will be 1.4 °C above pre-industrial levels by the end of 2023. Indeed, 2023 was the warmest year in 174 years of observational history, and the Earth has never been closer to the threshold of 1.5 °C above pre-industrial temperatures, which can irreversibly damage ecosystems and biodiversity [5]. Given the trend of global population growth and humanity’s increasing demand for food of animal origin, technologies, and worldwide transportation, limiting the further rise in global temperatures will only be achievable by mitigating GHG emissions in various sectors [6]. The agricultural sector has great potential to reduce GHG emissions and mitigate climate change as ruminant husbandry is responsible for 28% of total anthropogenic methane (CH_4_) emissions [7,8]. CH_4_ is seen as a “climate forcing” gas with a 27.9 times more potent global warming potential (GWP) compared to carbon dioxide (CO_2_) over a 100-year timescale [2,9]. Up to 90% of CH_4_ produced on ruminant farms is enteric CH_4_ originating from fermentation processes in the rumen, with dairy cows as significant contributors, while the rest (10%) is attributed to manure management [10,11]. Therefore, the target of most mitigation strategies is enteric CH_4_ in dairy cows, especially since its production is accompanied by a loss of gross energy intake estimated at 2–15% [12].

Nowadays, developed countries base their strategies to reduce enteric CH_4_ emissions in ruminant production, mainly on dietary supplements. Accordingly, numerous supplements have been tested for mitigating enteric CH_4_ in ruminants [13], and one of the novelties is introducing different seaweed species into the diet of ruminants [14]. The effectiveness of red seaweed in reducing enteric CH_4_ emissions was previously demonstrated [15] and explained by the high content of halogenated hydrocarbons (HHCs), such as bromomethane and bromoform, which inhibit methanogenesis [13]. In this regard, Kinley et al. [16] conducted an in vitro study and showed that the application of a 2% extract of the red seaweed *Asparagopsis taxiformis* reduced CH_4_ production to an almost undetectable level. Similarly, Machado et al. [17] decreased CH_4_ production in vitro by 95% with a 2% extract comprising *A. taxiformis*. The research was extended to in vivo conditions, and Kinley et al. [18] found that adding red seaweed *A. taxiformis* at 0.2% of TMR reduced CH_4_ production in cows by 98%. Despite the successful reduction in CH_4_ emissions with a dietary supplementation with red seaweeds, there are divided opinions on the expediency of their application because this process is accompanied by the release of HHCs into the atmosphere, which contributes to climate change in various ways [19,20]. Under these circumstances, brown seaweed supplementation trials have a slight advantage as they release a low amount of HHCs [19]. To the best of our knowledge, there are limited data on the effectiveness of brown seaweed in mitigating enteric CH_4_ emissions. The results of in vitro studies showcase that this is achievable [21,22]. However, it is necessary to examine the effect of supplements under in vivo conditions on the animal species in which they will be applied. This is important as attempts to reduce enteric CH_4_ emissions with dietary supplements, including brown seaweeds, should not impair ruminant productivity, health, and welfare in combating climate changes [23]. On the other hand, it is essential to examine how the CH_4_ mitigation measures based on dietary supplements affects the composition of the final product (milk or meat) due to the possible indirect effect on consumer health.

Therefore, this study aimed to investigate whether dietary supplementation with brown seaweed *A. nodosum* can reduce enteric CH_4_ emissions and to evaluate the effects of dietary treatment with *A. nodosum* on hematological and blood biochemical parameters, chemical composition of milk, and fatty acid profile in the peak-lactating Holstein cows.

## 2. Materials and Methods

### 2.1. Ethics Statement

The trial was conducted from March to May 2023 at a commercial dairy farm (Kovilovo, AlDahra Corporation, Belgrade, Serbia) (44°56′08.6″ N, 20°28′44.5″ E). The experimental protocol was evaluated and approved by the Veterinary Directorate of the Ministry of Agriculture, Forestry and Water Management of the Republic of Serbia (approval number 323-07-11720/2020-05/4) under the National Regulation on Animal Welfare.

### 2.2. Animals, Management, and Experimental Design

In the present study, 18 multiparous and clinically healthy Holstein-Friesian peak-lactating cows were selected and housed in the same barn with a one-week acclimation period as described by Muizelaar et al. [24]. Subsequently, the selected cows were randomly assigned to 1 of 3 numerically equal (*n* = 6) treatment groups: 0 mL/day of brown seaweed (*Ascophyllum nodosum*) supplemented with a total mixed ration (TMR) (control diet, CON); 50 mL/day of *A. nodosum* supplemented with a TMR (BS_50_); and 100 mL/day of *A. nodosum* supplemented with a TMR (BS_100_). Dietary treatment lasted 30 consecutive days, and the entire daily dose was mixed into the morning TMR. The brown seaweed product (SimbiWay B2^®^, Ozonway d.o.o., Belgrade, Serbia) was purchased from the same lot to minimize variation in nutritional content and had the following composition: *A. nodosum* (10%), carbohydrates (≤1%), ash (≤0.2%), protein (≤0.2%), cellulose (≤0.2%), and water (the rest). As the coarser particles of seaweed biomass tend to settle, the pack is stirred vigorously before each use to ensure homogeneity. All cows received the same TMR, which met or exceeded the National Research Council Requirements for Dairy Cattle [25] and was administered in equal portions twice daily at 6:00 am and 5:00 pm. The ingredient list, chemical composition, and nutritional value of the TMR are provided in Table 1. Drinking water was available ad libitum. Cows were milked three times daily at 5:00 am, 12:00 am, and 7:00 pm. The health status of the cows was checked daily by farm veterinarians and researchers during this study, and no clinical signs of disease were detected. Air temperature (T) and relative humidity (RH) were measured daily using an electronic device (Lutron LM-8010, Lutron Electronic Enterprise Co., Ltd., Taipei, Taiwan) and were in the following range: minimum (T = 15.7 °C; RH = 38.8%), maximum (T = 23.7 °C; RH = 59.2%), and average (T = 18.9 °C; RH = 52.1%). 

### 2.3. Sample Collection and Measurement Scheme

In this study, there were three measurement and sampling weeks: 1 week before the start of the dietary treatment (period 1, P1); 1 week after 15 days of metabolic adaptation—middle of the dietary treatment (period 2, P2); and one week after the end of the 30-day dietary treatment (period 3, P3). During each measurement and sampling period, the procedures were performed in the following order and duration: dry matter intake and milk yield were recorded every other day to obtain a reliable average per period, enteric methane (CH_4_) emissions were measured for four consecutive days to obtain a reliable estimation per period, and venous blood and milk samples were collected once per period. 

### 2.4. Determination of Dry Matter Intake and Milk Yield

Dry matter intake (DMI) was determined by weighing the amount of feed allocated and feed orts by measuring the dry matter content of meal and orts as described by Thorsteinsson et al. [26]. As part of regular farm practice, milk yield was recorded using an electronic device connected to the milking system. Milk yield was also expressed as kilograms of fat/protein-corrected milk (FPCM) following the equation proposed by the International Dairy Federation [27]: kg FPCM = Milk kg × ((0.1226 × Fat %) + (0.0776 × Protein %) + 0.2534)

### 2.5. Measurement of Enteric CH_4_ Emissions and Processing Data Obtained 

Enteric CH_4_ emissions were measured twice daily (2 to 4 and 6 to 8 h after morning feeding) for four consecutive days in each cow using a hand-held laser methane detector (LMD Mini-Green; Tokyo Gas Engineering Solutions, Tokyo, Japan). The laser beam was pointed into the nostrils of the animals to determine the CH_4_ concentration in exhaled air. A single measurement continuously lasted 4 min per cow, and LMD was set for recording CH_4_ concentrations in narrow intervals of 0.5 s. Thus, each measurement of enteric CH_4_ emissions resulted in a time series of CH_4_ concentrations comprising 480 values belonging to a single animal marked as a CH_4_ concentration profile. LMD was used to measure the concentration of CH_4_ concentrations in ppm × meter; however, since the distance between the nostrils of the animals and LMD was 1 m, all values were expressed in ppm, which was similar to the procedure carried out by Pinto et al. [28]. At the beginning of each measurement session, the LMD was connected via Bluetooth to the operator’s cell phone, and the Leak Finder application was run to export and store the data obtained. The same operator measured enteric CH_4_ concentrations throughout the experiment to minimize the possible influence of this factor on the measurement results. All measurements were performed on standing animals and in a windless environment. LMD data processing involved the inspection of each CH_4_ profile. The lowest CH_4_ value in the profile was considered a background concentration and was subtracted from all other individual values of the respective data set, as described by Sorg [29]. The average of all CH_4_ values was chosen in this study to compare enteric CH_4_ emissions’ phenotypes among the examined groups of cows, as described by Grobler et al. [30] and Niero et al. [31]. Finally, the three parameters of CH_4_ emissions data were derived in this study, including CH_4_ production (CH_4_, ppm), CH_4_ yield (CH_4_/DMI, ppm/kg), and CH_4_ intensity per kg of FPCM (CH_4_/FPCM, ppm/kg), as proposed by Grešáková et al. [32]. 

### 2.6. Collection and Analysis of Blood Samples

Blood samples were drawn from the jugular vein using an 18-gauge needle and were transferred into vacutainer tubes containing lithium-heparin (6.0 mL, BD Vacutainer, Plymouth, Devon, UK) for hematological analyses and into vacutainer tubes containing a clot activator (10.0 mL, BD Vacutainer, Plymouth, Devon, UK) for biochemical analyses. The sampling procedure was performed approximately 1 h before feeding in the morning, and the samples were placed in an icebox and transferred to a laboratory within 30 min. Analyses of hematological parameters were performed immediately after the samples were received in the laboratory, while the samples for the analyses of biochemical variables were allowed to clot spontaneously. After clotting, samples were centrifuged at 1500× *g* for 10 min to separate the serum, were decanted into graduated polypropylene tubes (1.5 mL, Eppendorf AG, Hamburg, Germany), and were then stored at −20 °C until analysis.

Hematological parameters, including red blood cell count (RBC, 10^12^/L), hemoglobin concentration (g/dL), hematocrit (%), mean corpuscular volume (MCV, fL), mean corpuscular hemoglobin (MCH, pg), mean corpuscular hemoglobin concentration (MCHC, g/dL), total white blood cell (WBC, 10^9^/L) count, granulocyte (10^9^/L) count, monocyte (10^9^/L), and lymphocyte (10^9^/L) count, were determined automatically using a three-part differential hematology analyzer (Phoenix NCC-30Vet, NeoMedica, Belgrade, Serbia).

On the other hand, analyses of biochemical indicators included the determination of total protein (g/L), albumin (g/L), blood urea nitrogen (BUN; mmol/L), total bilirubin (μmol/L), gamma-glutamyltransferase (γ-GT), aspartate aminotransferase (AST; U/L), triacylglycerols (TAG; mmol/L), total cholesterol (mmol/L), HDL-cholesterol (HDL-C; mmol/L), glucose (mmol/L), beta-hydroxybutyrate (BHB; mmol/L), and non-esterified fatty acids (NEFA; mmol/L). Biochemical indicators were analyzed using the respective methods/kits: total protein (biuret reaction); albumin (bromcresol green method), BUN (urease/glutamate dehydrogenase method); total bilirubin (diazotized sulphanilic acid method); γ-GT and AST (IFCC method); TAG (glycerol phosphate oxidase/peroxidase); total cholesterol (cholesterol oxidase/peroxidase method), HDL-C (direct detergent method), and BHB (enzymatic method) by BioSystems SA (Barcelona, Spain); and NEFA (colorimetric method) by Randox Laboratories Ltd. (Crumlin, Northern Ireland, UK). Analyses were performed automatically with a biochemical analyzer (BioSystem A15, BioSystems SA, Barcelona, Spain). Glucose was measured in whole blood enzymatically (glucose dehydrogenase, GDH-NAD method) using commercial test strips (Abbott Diabetes CareLtd., Oxon, UK). 

### 2.7. Collection and Analysis of Milk Samples

Milk samples were collected with a milk meter (DeLaval MM6, DeLaval Ltd., Gurnee, IL, USA) in plastic bottles during milking in the morning with a total volume of 100 mL for each cow. Of this amount, 90.0 mL was intended for chemical composition analyses, 8.0 mL was sub-sampled into plastic tubes for the determination of total somatic cell count (SCC), and 2.0 mL was placed into graduated polypropylene tubes (Eppendorf AG, Hamburg, Germany) for the determination of fatty acids composition. Analyses of the chemical composition of milk included the determination of protein (%), fat (%), lactose (%), total solids (%), and solid non-fat (%) content with an ultrasonic method using a Lactoscan (Milkotronik Ltd., Stara Zagora, Bulgaria). Total SCC was determined optically using an automated device (DeLaval Cell Counter-DCC, DeLaval Ltd., Gurnee, IL, USA). The composition of milk fatty acids was analyzed using the gas chromatography-flame ionization detection (GC-FID) method. Namely, 2.0 mL of a milk sample in polypropylene graduated microtubes (Eppendorf AG, Hamburg, Germany) was centrifuged at 10,000 rpm for 30 min at 4 °C. The upper lipid layer (100 to 150 mg) was removed and placed into a 10.0 mL glass cuvette for direct extraction and underwent acid-catalyzed trans-methylation with 1.5 mL of 3M HCl in methanol to allow us to obtain fatty acid methyl esters (FAMEs). The mixture was vortexed and heated in a water bath at 85 °C for 45 min and then cooled, and hexane (Sigma Aldrich, Burlington, MA, USA) was added for FAME extraction according to the procedure described by Ichihara and Fukubayashi [33]. Following centrifugation for 10 min at 3000 rpm, the upper hexane layer containing the FAMEs was transferred into vials and immediately analyzed. Fatty acid methyl esters were determined using gas chromatography Agilent technologies (Santa Clara, CA, USA) AGILENT 7890 GC Chem StationOpeartion with an FID detector. The FAMEs were separated on a capillary column (CP-Sil88, 100 m × 0.25 mm, 0.2 μm film thickness; SUPELCO, Bellefonte, PA, USA) under the following conditions: 1 μL injections of the FAME mixture were made in split mode 20:1; the injector temperature was 250 °C with an injector split flow of 20 mL/min, a pressure of 31,623 psi, and a total flow of 24 mL/min; the oven temperature program started at 80 °C, increasing by 4 °C/min up to 220 °C (hold time 5 min), then by 4 °C/min up to 240 °C, and was then held at 240 °C for 10 min; the carrier gas (He) flow rate was adjusted to 1.0 mL/min and makeup gas-nitrogen flow to 25 mL/min; and FID detector temperature was 270 °C. The run time was 55 min. Chromatographic peaks were identified by comparing their retention times with the standard FAME mix (Supelco FAME Mix, Bellefonte, PA, USA). The quantification was based on the ratio between all peak areas and the corresponding peak. The results were expressed as a percentage (%) of individual fatty acids of all determined fatty acids.

### 2.8. Data Analysis

The data were statistically analyzed using SPSS Statistics 29.0 software (IBM^®^, Armonk, NY, USA). The normality assumption was checked with the Shapiro–Wilk test, and only the milk SCC had to be logarithmically transformed (Log_10_ SCC). To statistically test the influence of the dietary treatment, period, and its interaction with the examined parameters, analysis of variance (ANOVA) was applied with dietary treatment as a between-subject factor, period as a within-subject factor, and with the examined parameter as a dependent variable. The results obtained one week before supplementation (P1) were included in the model as covariates. The Bonferroni procedure was used for pairwise comparisons. All data were expressed as mean ± standard error (SE). Significance was declared at *p* < 0.05, *p* < 0.01 and *p* < 0.001, and a tendency was acknowledged at 0.05 ≤ *p* < 0.10.

## 3. Results

There were no significant differences (*p* > 0.05) between the groups of cows formed at the start of this study in terms of average (±standard error) parity, days in milk, milk yield, body weight, and body condition score (Appendix A), which indicates that further comparisons throughout this study warrant uniformity and relevance.

### 3.1. Enteric CH_4_ Emissions and Production Performance Parameters

There was a significant effect of dietary treatment on CH_4_ production (*p* < 0.001), CH_4_ yield (*p* < 0.001), CH_4_ intensity (*p* = 0.002), and DMI (*p* = 0.025). In addition, a significant effect of the treatment × period interaction on CH_4_ production (*p* = 0.041) and CH_4_ yield (*p* = 0.023) was found (Table 2). Therefore, seaweed supplementation significantly reduced (*p* < 0.05) CH_4_ production and CH_4_ yield in both BS_50_ and BS_100_ cows compared to CON cows. This effect is noticeable in P2, but also in P3, when the lowest values for these parameters were recorded in the seaweed-supplemented cows. Similarly, CH_4_ intensity was significantly reduced (*p* < 0.05) in BS_50_ cows compared to CON cows already in P2; however, in P3, both groups of seaweed-supplemented cows had significantly lower CH_4_ intensity (*p* < 0.05) compared to CON cows. DMI was significantly higher (*p* < 0.05) in BS_50_ than in CON cows only in P2. No significant effects (*p* > 0.05) of treatment, period, and treatment × period interaction on FPCM and MY were found.

### 3.2. Hematological and Biochemical Parameters

In the present study, treatment, period, and treatment–period interaction had no significant effect (*p* > 0.05) on the hematological parameters. The only exception is hemoglobin, which was significantly affected by period (*p* = 0.032). It was observed that CON and BS_50_ cows displayed a significant decrease (*p* < 0.05) in hemoglobin concentration from P2 to P3. A similar decline was noted from P2 to P3 in RBC (*p* < 0.05), but only within BS_50_ cows (Table 3).

There were no significant effects (*p* > 0.05) of dietary treatment, period, and treatment–period interaction on most blood biochemical parameters. However, dietary treatment tended to affect the concentrations of total bilirubin (*p* = 0.060) and TAG (*p* = 0.079). In addition, TAG was significantly affected by the dietary treatment × period interaction (*p* = 0.018) (Table 4). In this regard, the concentration of total bilirubin was significantly higher in BS_50_ cows compared to both CON (*p* < 0.05) and BS_100_ (*p* < 0.05) cows in P2, while TAG concentration was significantly lower (*p* < 0.05) in BS_50_ cows compared only to CON cows in P3. 

### 3.3. Chemical Composition, Fatty Acid Profile, and Somatic Cell Count of the Milk

No significant effects (*p* > 0.05) of dietary treatment, period, and treatment × period interaction on the chemical composition of milk were recorded. However, pairwise comparisons showed that BS_50_ cows had a significantly higher (*p* < 0.05) milk fat content than CON cows in P3 (Table 5).

Regarding the milk fatty acid profile, the results showed that dietary treatment had a significant effect on the proportions of C16:0 (*p* = 0.023) and C18:3 n-3 (*p* = 0.007), while the proportions of C4:0 (*p* = 0.085) and C17:0 (*p* = 0.057) showed similar tendencies (Table 5). Thus, pairwise comparisons showed that both BS_50_ (*p* < 0.05) and BS_100_ (*p* < 0.05) cows had significantly higher proportions of 16:0 than CON cows in P3. In addition, BS_100_ cows had a significantly lower (*p* < 0.05) proportion of 17:0 only compared to CON cows in P3 and significantly higher (*p* < 0.05) proportions of C18:3 n3 in comparison to both CON and BS_50_ cows in P3. 

There was no effect of period (*p* > 0.05) on the proportion of any milk fatty acid, but the proportions of C8:0 (*p* = 0.021), C10:0 (*p* = 0.039), C12:0 (*p* = 0.010), and C16:1 (*p* = 0.031) were significantly affected by the dietary treatment × period interaction (Table 5). In this regard, BS_50_ and BS_100_ cows had numerically lower proportions of C8:0, C10:0, and C12:0 compared to CON cows in P2. On the other hand, the proportion of C16:1 was significantly higher (*p* < 0.05) in BS_50_ cows compared to CON cows in P3. 

Finally, the milk somatic cell count (Log_10_ SCC) was not affected by treatment (*p* = 0.248), period (*p* = 0.936), or the interaction of treatment and period (*p* = 0.807).

## 4. Discussion

The results of the present study are the first to show that dietary treatment with brown seaweed *A. nodosum* (10%) in daily doses of 50 mL and 100 mL can successfully reduce enteric CH_4_ emissions in peak-lactating dairy cows. This effect was achieved without impairing cows’ productivity and health but rather by improving their metabolic status, chemical composition of their milk, and their fatty acid profile.

In this regard, ANOVA indicated that the supplementation with brown seaweed *A. nodosum* significantly reduced enteric CH_4_ production in peak-lactating dairy cows. The results also suggest that the interaction between dietary treatment and period significantly affect enteric CH_4_ production (Table 2). Thus, a significant decrease in enteric CH_4_ production by 34.5% and 23.5% was recorded after 15 days of supplementation in both BS_50_ and BS_100_ cows, respectively, compared to the initial measurements that preceded supplementation with *A. nodosum*. In addition, at the end of the 30-day *A. nodosum* supplementation trial, an overall decrease in enteric CH_4_ emissions of 43.4% and 44.5% was observed in BS_50_ and BS_100_, respectively, compared to the pre-supplementation period. Moreover, enteric CH_4_ emissions were significantly lower in BS_50_ and BS_100_ compared to the CON group of cows after 15 days (P2) and 30 days (P3) of supplementation. The results reveal that supplementation with *A. nodosum* significantly reduces the emissions of enteric CH_4_ when expressed per kilogram of dry matter intake (DMI) (CH_4_ yield) and kilogram of fat/protein-corrected milk (FPCM) (CH_4_ intensity). Pairwise comparisons did not show differences in any enteric CH_4_ emission parameters, including CH_4_ production, CH_4_ yield, and CH_4_ intensity, between BS_50_ and BS_100_ cows in P2 and P3 (Table 2). These can be crucial data in considering the economic aspects of strategies to mitigate CH_4_ from ruminant husbandry because the desired effects can be achieved even with a lower supplement dose. 

The present study was preceded by the results published by Ramin et al. [21], who documented that another species of brown seaweed, *Alaria esculenta*, significantly reduces in vitro CH_4_ emissions. This effect of brown seaweed, including the species *A. nodosum*, on enteric CH_4_ emissions can be related to their chemical composition. Namely, brown seaweed contains polyphenolic compounds, the most important of which are phlorotannins, which can inhibit the growth of fibrolytic microbes in the rumen, such as *Fibrobacter succinogenes* [18]. It is known that the size of the population of fibrolytic microbes in the rumen correlates with the production of enteric CH_4_ since the metabolism of these bacteria is accompanied by the release of hydrogen (H_2_), thus becoming available to methanogens for CH_4_ synthesis [34]. The described cooperation between fibrolytic and methanogenic microbes is known as interspecies H_2_ transfer and represents a syntrophic relationship between two microbes [35]. In addition, the reduction in enteric CH_4_ emissions, achieved by applying *A. nodosum* in our study, can be related to the decrease in digestibility in the rumen, which was previously reported by Gemeda et al. [36]. It would be interesting to refer to the results published by Antaya et al. [37], who administered the brown seaweed *A. nodosum* to Jersey cows but which had no effect on enteric CH_4_ emissions. The differences in the results obtained in our study and the study conducted by Antaya et al. [37] should initially relate to the content of *A. nodosum* in commercial seaweed products, the daily dose, and the duration of supplement use. Namely, Antaya et al. [37] applied the supplement ten days earlier than in our study, and the content of *A. nodosum* in the supplement used by these authors is unknown. Furthermore, Antaya et al. [37] used the GreenFeed system to measure enteric CH_4_ emissions, while an LMD was applied in our study. Although it is easier to standardize the number and period of measurements per animal when applying LMD, the GreenFeed system has greater repeatability compared to the LMD [29]. Due to the comparative advantages and disadvantages of these methods, the difference in the obtained results between the study of Antaya et al. [37] and our study can be attributed to the applied methodology. Therefore, it should be pointed out that we obtained these results in the experimental design, form, and length of application of the brown seaweed (*A. nodosum*) product, as well as the chosen methodology for measuring enteric CH_4_ emission (LMD), which are described in the presented study.

In contrast to our study, Thorsteinsson et al. [26] observed no reduction in CH_4_ emissions by a separate administration of three species of brown seaweeds, including *A. nodosum*, *Saccharina latissima*, and *Sargassum muticum*, in Danish Holstein dairy cows in mid-lactation. However, these authors also conducted in vitro tests, and they found that *A. nodosum* reduces CH_4_ production. The lack of effect of *A. nodosum* on CH_4_ production in cows in the study performed by Thorsteinsson et al. [26] could be speculatively explained by differences in the formulation and properties of the *A. nodosum* commercial product. Indeed, pre-processing in the manufacture of seaweed products includes drying, which, depending on temperature and duration, can influence the content of phlorotannins and other antimethanogenic components in the product [38]. However, it is not yet known how the different types of pre-processing of seaweed affect the desired effects of the product under both in vitro and in vivo conditions. In addition, the period of dietary treatment in the study by Thorsteinsson et al. [26] lasted 14 days. It remains an open question what the effects would be if the treatments lasted longer (at least 30 days) because our results indicate that the reduction of CH_4_ production exists after 15 days of supplementation (up to 30 days of supplementation).

Concerning the impact of the brown seaweed *A. nodosum* on the reduction in enteric CH_4_ production achieved in our research, it is essential to note that it did not affect the productivity of the examined cows. This is crucial for applying nutritional strategies to reduce CH_4_ emissions from cattle farms [39]. These views are supported by the results of ANOVA, showing no significant effect of *A. nodosum* supplementation on FPCM and MY in the present study. Although the treatment had no significant effect on FPCM and MY, it is interesting that BS_50_ cows had numerically higher values for both parameters than CON and BS_100_ cows. Perhaps this can be explained by a significantly higher DMI in P2 and a numerically higher DMI in P3 in BS_50_ cows compared to CON and BS_100_ cows due to the positive effect of DMI on cow productivity [40]. Similarly, Antaya et al. [36] also observed an increased DMI in cows supplemented with *A. nodosum*. On the other hand, Newton et al. [41] and Thorsteinsson et al. [26] found no such effect of *A. nodosum* on DMI. Therefore, the impact of *A. nodosum* on DMI is inconsistent, and it is still unknown why the effect of increasing DMI in cows supplemented with *A. nodosum* is absent in some cases.

Hematological analysis is a valuable tool in bovine medicine for assessing the general health status of cows and diagnosing various disorders [42]. In this context, ANOVA did not detect any significant effect of dietary treatment with *A. nodosum* on red and white blood cell parameters. Similarly, Newton et al. [41] reported no impact of *A. nodosum* supplementation on hematologic parameters in Holstein dairy cows. All hematological parameters examined in the present study were within the reference range given by Wood and Quiroz-Rocha [43]. 

The results of the blood biochemical parameters in the present study indicate that *A. nodosum* supplementation has no negative implications on the metabolic status of peak-lactating dairy cows, as there was no significant effect of the treatment on total protein, albumin or blood urea nitrogen, total and HDL-cholesterol, glucose and beta-hydroxybutyrate (BHB) concentrations, and on gamma-glutamyltransferase (γ-GT) and aspartate-aminotransferase (AST) activities. However, in our study, ANOVA revealed that *A. nodosum* supplementation tended to affect the serum concentrations of triacylglycerols (*p* = 0.079) and total bilirubin (*p* = 0.060). Thus, serum triacylglycerols concentrations were significantly lower in BS_50_ cows and numerically lower in BS_100_ cows at the end of the 30-day supplementation period (P3) than in CON cows, indicating a more favorable energy status of cows receiving *A. nodosum* in their diet. Namely, cows with intense lipomobilization and hyperketonemia are known to experience changes in lipid metabolism, which is reflected by increased serum triacylglycerols concentrations [44]. Furthermore, multivariate analyses showed that serum triacylglycerols concentrations can be predictive biomarkers for hyperketonemia. Accordingly, higher triacylglycerols concentrations indicate a more pronounced negative energy balance (NEB), which is associated with more extensive fat mobilization and an increased synthesis of triacylglycerols in the liver via the re-esterification of non-esterified fatty acids (NEFA) [45]. 

The changes and differences in total bilirubin concentration during supplementation with the brown seaweed *A. nodosum* were not uniform or regular. The highest total bilirubin concentrations were recorded in BS_50_ cows at the end of the supplementation period (P3). Such results indirectly indicate that higher total bilirubin concentrations were found in the cows with lower enteric CH_4_ emissions. These findings can be linked to the results of Bošnjaković et al. [46], who showed a significant negative correlation between enteric CH_4_ emissions and serum concentration of total bilirubin. However, how these parameters are related is not yet known, and further studies are needed to clarify the nature of their interdependence. All blood biochemical parameters investigated in the present study were within the reference intervals described for dairy cows [47,48].

No parameters of the chemical composition of milk were affected by the dietary treatment with *A. nodosum* in our study. Similar results were reported by Antaya et al. [36], who administered *A. nodosum* to Jersey cows. Newton et al. [41] and Thorsteinsson et al. [26] also observed no changes in the chemical composition of milk after supplementation with *A. nodosum* in mid-lactating Holstein cows. However, pairwise comparisons in our study showed a significantly higher milk fat content in BS_50_ than in CON cows in P3. This is one of the few cases of increased milk fat content in cows treated with brown seaweed in the available literature. To the best of the authors’ knowledge, only Xue et al. [49] have obtained such results, especially for milk fat content, by supplementing cows with brown seaweed *Thallus laminariae*. In addition, these authors also investigated the rumen metabolome of supplemented cows. They explained the higher content of milk fat by a higher proportion of ruminal acetate as a precursor for milk fat synthesis, which can also be an applicable explanation to our study. It is necessary to further investigate the rumen metabolome in cows supplemented with *A. nodosum* to resolve the findings discussed above. It is important to investigate the long-term effects of *A. nodosum* supplementation on milk composition, especially if the application of this supplement would be part of a regular mitigation strategy in the future. 

The proportion of milk fatty acids is interesting to consider and discuss not only from the perspective of the productivity of the cows studied but also from the perspective of human health due to its possible association with various cardiovascular, endocrinological, and oncological diseases [50,51]. In this regard, the results of our study clearly showed that dietary treatment with *A. nodosum* significantly affects the proportions of C16:0 and C18:3 n-3. At the same time, C4:0 (*p* = 0.085) and C17:0 (*p* = 0.057) tended to be affected via treatment (Table 5). The results of the pairwise comparisons showed a significantly lower proportion of C4:0 in BS_50_ cows compared to CON and BS_100_ cows in P2. There are currently no comparable studies that documented this finding. The proportion of C4:0 differed between the two groups of cows fed brown seaweed. Possible explanations should be sought at the level of the mammary gland, as C4:0 originates from de novo synthesis [51]. It is worth mentioning that even a low content of C4:0 has a health benefit for consumers because it inhibits the growth of various human cancers, primarily colon cancer [50]. The responsible mechanisms are described as the influence of C4:0 on histone hyperacetylation and the suppression of genes for cell growth [52], the modulation of the expression of genes related to oxidative and metabolic stresses [53], and the induction of cell apoptosis [54]. The proportion of C16:0 was significantly higher in both BS_50_ and BS_100_ compared to CON in P3. C16:0 has been recognized as a hypercholesterolemic fatty acid that may contribute to the development of cardiovascular diseases [55]. However, the study by Thorning et al. [56] provided new insights into the effect of this fatty acid from dairy sources on consumers’ health since no harmful effects on cardiovascular parameters were found. The fate of C16:0 in humans is also influenced by carbohydrate intake, and the importance of C16:0 can hardly be viewed separately from other factors in the human diet [57]. Our results also showed a significantly higher proportion of C16:1 in BS_50_ cows compared to CON cows in P3. This may be a beneficial finding for consumers due to the potential preventive effect of C16:1 on insulin resistance and diabetes [58]. The proportion of C17:0 was significantly lower in BS_100_ cows than in CON cows in P3, while BS_50_ had a numerically lower content of this fatty acid than CON in the same experiment period. Similar to our study, Lopez et al. [59] reported a lower proportion of C17:0 after dietary supplementation with *A. nodosum* in dairy cows. However, the results in Table 5 also show that the proportion of C17:0 was lower in P1 (before supplementation) in both BS_50_ and BS_100_ than in CON cows, which may indicate that the proportion of this fatty acid at the end of the 30-day supplementation (P3) period with *A. nodosum* was not only an effect of the dietary treatment. Moreover, these results can be explained by a genetic effect because the herd effect was minimized due to the uniform management of cows examined in this study [60]. Pairwise comparisons revealed that C18:3n-3 proportions were significantly higher in BS_100_ cows than in BS_50_ and CON cows in P3. This is a favorable finding for consumer health since C18:3n-3 has anti-inflammatory, anti-cancer, and anti-atherogenic properties [58]. In addition to the public health context, milk fatty acids can also be seen as potential predictors of CH_4_ emissions in cows. In our study, we presented and discussed milk fatty acids that were affected by brown seaweed dietary treatment. Further studies should reveal whether the aforementioned fatty acids are valuable in predicting enteric CH_4_ emissions. Although no significant treatment effect was found regarding the milk somatic cell count (SCC), it is essential to note that both groups of cows supplemented with brown seaweed had a numerically lower SCC, suggesting a more favorable health quality of milk. Lopez et al. [59] explained a similar result with the positive effects of *A. nodosum* on the supplemented cows’ immune function and antioxidant capacity.

## 5. Conclusions

In conclusion, the results of this study show that supplementation with the brown seaweed *A. nodosum* successfully reduces enteric CH_4_ emissions in peak-lactating dairy cows. No differences in the effects on CH_4_ emissions were observed between cows supplemented with lower and higher doses of *A. nodosum*, which may be an important factor in considering economic aspects when formulating comprehensive strategies to reduce the release of CH_4_ in ruminant production. Furthermore, dietary treatment with *A. nodosum* did not affect cow health and productivity. In addition, the findings of the present study indicate the possible potential of the dietary inclusion of *A. nodosum* to increase DMI but also to induce some favorable changes in energy status and milk composition, especially in the fatty acid profile of peak-lactating dairy cows. However, further studies are needed to explain the underlying mechanisms that lead to the described treatment results with *A. nodosum* in cows, especially concerning the rumen metabolome and microbiome, including proteomic and lipidomic studies of the mammary gland.

## Figures and Tables

**Table 1 animals-14-01520-t001:** Composition and nutritional value of the total mixed ration for peak-lactating cows in this study.

Ingredients, % of DM/Day	Content
Corn silage	29.9
Alfalfa haylage	6.9
Alfalfa hay	11.4
Brewer’s grain (wet)	5.2
Molasses	5.0
Cottonseed meal	4.0
Palm oil	2.2
Corn grain	20.0
Barley	1.7
Rye grain	0.7
Wheat grain	1.1
Sunflower meal (34%CP)	16.0
Sodium bicarbonate	0.1
Calcium carbonate	0.7
NaCl	0.5
Monocalcium phosphate	0.1
Vitamin Mineral Mix	0.5
Nutritional value	
Dry matter (%)	51.30
Ash % of DM	7.80
Fat, % of DM	6.93
Starch, % of DM	20.40
Sugar, % of DM	7.30
Crude protein, % of DM	16.70
RDP, % of DM	12.13
RUP, % of DM	4.56
MP (g/kg)	91.34
NDF (%)	27.60
ADF (%)	18.60
Ca (%)	1.17
P (%)	0.50
Mg (%)	0.27
K (%)	1.35
S (%)	0.22
Na (%)	0.21
Cl (%)	0.47
Fe (ppm)	303.00
Mn (ppm)	68.00
Zn (ppm)	81.00
Cu (ppm)	20.00
Energy value—metabolic energy (MJ/kg DM.)	12.38
DCAD (meq/kg DM)	167.00

RDP—rumen degradable protein; RUP—rumen undegradable protein; MP—metabolizable protein; NDF—neutral detergent fiber; ADF—acid detergent fiber; DCAD—dietary cation-anion difference.

**Table 2 animals-14-01520-t002:** Mean ± SE and ANOVA *p*-values for the effects of brown seaweed (10% *Ascophyllum nodosum*) supplementation on enteric CH_4_ emission parameters, dry matter intake, fat/protein-corrected milk, and milk yield.

Parameter (Unit)	Group ^1^	Period of the Experiment ^2^			*p*-Value ^3^		
		P1	P2	P3	T	P	T × P
CH_4_ production (ppm)	CON	129.6 ± 4.9	124.1 ± 4.1 ^Aa^	94.5 ± 3.7 ^Ba^	<0.001	0.878	0.041
	BS_50_	121.3 ± 9.3	79.4 ± 3.3 ^Ab^	68.6 ± 2.1 ^Bb^			
	BS_100_	118.5 ± 2.3	90.6 ± 6.1 ^Ab^	65.8 ± 3.2 ^Bb^			
CH_4_ yield (ppm/kg DMI)	CON	6.1 ± 0.2	6.0 ± 0.2 ^Aa^	4.5 ± 0.2 ^Ba^	<0.001	0.607	0.023
	BS_50_	5.6 ± 0.4	3.7 ± 0.2 ^Ab^	3.2 ± 0.2 ^Bb^			
	BS_100_	5.6 ± 0.1	4.4 ± 0.3 ^Ab^	3.1 ± 0.1 ^Bb^			
CH_4_ intensity (ppm/kg FPCM)	CON	3.4 ± 0.1	3.3 ± 0.2 ^Aa^	2.7 ± 0.2 ^Ba^	0.002	0.720	0.325
	BS_50_	3.1 ± 0.3	2.0 ± 0.1 ^Bb^	1.9 ± 0.2 ^Bb^			
	BS_100_	3.2 ± 0.2	2.6 ± 0.3 ^Aab^	1.9 ± 0.1 ^Bb^			
DMI (kg/day)	CON	21.3 ± 0.2	20.6 ± 0.5 ^Aa^	20.9 ± 0.2 ^Aa^	0.025	0.085	0.986
	BS_50_	21.5 ± 0.1	21.6 ± 0.1 ^Ab^	21.2 ± 0.5 ^Aa^			
	BS_100_	21.1 ± 0.2	20.9 ± 0.2 ^Aa^	20.9 ± 0.3 ^Aa^			
FPCM (kg/day)	CON	38.6 ± 1.5	38.0 ± 2.0 ^Aa^	35.1 ± 2.1 ^Aa^	0.762	0.763	0.639
	BS_50_	40.2 ± 1.7	39.3 ± 1.1 ^Aa^	37.6 ± 1.4 ^Aa^			
	BS_100_	37.9 ± 1.4	35.9 ± 2.5 ^Aa^	35.8 ± 2.7 ^Aa^			
Milk yield (kg/day)	CON	41.4 ± 1.4	42.1 ± 2.6 ^Aa^	38.8 ± 2.1 ^Aa^	0.797	0.882	0.920
	BS_50_	42.6 ± 2.2	42.4 ± 1.5 ^Aa^	39.3 ± 1.7 ^Aa^			
	BS_100_	40.2 ± 0.9	39.4 ± 2.3 ^Aa^	37.1 ± 1.7 ^Aa^			

^1^ CON—control group of cows; BS_50_—group of cows supplemented with 50 mL of brown seaweed (10% *A. nodosum*, 10%); BS_100_—group of cows supplemented with 100 mL of brown seaweed (10% *A. nodosum*); ^2^ P1—the one-week period before brown seaweed supplementation; P2—the one-week period after 15 days of brown seaweed supplementation; P3—the one-week period after a total of 30 days of brown seaweed supplementation; ^3^ significance was declared at *p* < 0.05; T—treatment; P—period; T × P—interaction between treatment and period; DMI—dry matter intake; FPCM—fat/protein-corrected milk; ^AB^ different uppercase letters indicate statistically significant differences (*p* < 0.05) within the same group at different periods; ^ab^ different lowercase letters indicate statistically significant differences (*p* < 0.05) between groups at the same period.

**Table 3 animals-14-01520-t003:** Mean ± SE and ANOVA *p*-values for the effects of brown seaweed (10% *Ascophyllum nodosum*) supplementation on hematological parameters.

Parameter (Unit)	Group ^1^	Period of the Experiment ^2^			*p*-Value ^3^		
		P1	P2	P3	T	P	T × P
RBC (10^12^/L)	CON	5.4 ± 0.2	5.6 ± 0.3 ^Aa^	5.3 ± 0.2 ^Aa^	0.202	0.693	0.318
	BS_50_	5.4 ± 0.3	5.9 ± 0.2 ^Aa^	5.4 ± 0.1 ^Ba^			
	BS_100_	5.0 ± 0.3	5.2 ± 0.1 ^Aa^	5.1 ± 0.2 ^Aa^			
Hemoglobin (g/dL)	CON	9.7 ± 0.2	10.2 ± 0.3 ^Aa^	9.5 ± 0.4 ^Ba^	0.306	0.032	0.208
	BS_50_	10.0 ± 0.3	10.9 ± 0.1 ^Aa^	10.1 ± 0.2 ^Ba^			
	BS_100_	9.7 ± 0.3	10.1 ± 0.3 ^Aa^	9.7 ± 0.3 ^Aa^			
Hematocrit (%)	CON	24.4 ± 0.7	25.0 ± 0.8 ^Aa^	23.4 ± 0.9 ^Aa^	0.315	0.157	0.546
	BS_50_	24.3 ± 0.9	26.8 ± 0.4 ^Aa^	24.3 ± 0.8 ^Aa^			
	BS_100_	24.1 ± 1.1	25.1 ± 0.8 ^Aa^	24.1 ± 1.0 ^Aa^			
MCV (fL)	CON	45.1 ± 0.8	44.9 ± 0.7 ^Aa^	44.6 ± 0.9 ^Aa^	0.595	0.959	0.361
	BS_50_	45.8 ± 1.7	45.8 ± 1.4 ^Aa^	45.3 ± 1.4 ^Aa^			
	BS_100_	48.0 ± 1.2	48.7 ± 1.5 ^Aa^	47.1 ± 1.2 ^Aa^			
MCH (pg)	CON	17.9 ± 0.3	18.2 ± 0.4 ^Aa^	18.0 ± 0.4 ^Aa^	0.799	0.300	0.767
	BS_50_	18.8 ± 0.8	18.7 ± 0.8 ^Aa^	18.7 ± 0.5 ^Aa^			
	BS_100_	19.3 ± 0.6	19.5 ± 0.6 ^Aa^	18.9 ± 0.4 ^Aa^			
MCHC (g/dL)	CON	39.9 ± 0.3	40.7 ± 0.6 ^Aa^	40.6 ± 0.5 ^Aa^	0.377	0.620	0.967
	BS_50_	41.2 ± 0.6	40.8 ± 0.8 ^Aa^	41.5 ± 0.9 ^Aa^			
	BS_100_	40.3 ± 0.9	40.2 ± 0.8 ^Aa^	40.3 ± 0.3 ^Aa^			
WBC (10^9^/L)	CON	10.5 ± 0.5	13.5 ± 0.6 ^Aa^	11.0 ± 0.3 ^Aa^	0.302	0.877	0.206
	BS_50_	10.5 ± 0.3	12.1 ± 0.4 ^Aa^	10.5 ± 0.1 ^Aa^			
	BS_100_	9.4 ± 0.1	9.1 ± 0.2 ^Aa^	9.6 ± 0.1 ^Aa^			
Granulocytes (10^9^/L)	CON	3.0 ± 0.2	4.0 ± 0.1 ^Aa^	3.1 ± 0.2 ^Aa^	0.520	0.984	0.169
	BS_50_	2.6 ± 0.1	2.6 ± 0.3 ^Aa^	3.3 ± 0.1 ^Aa^			
	BS_100_	2.8 ± 0.2	3.3 ± 0.1 ^Aa^	3.8 ± 0.4 ^Aa^			
Monocytes (10^9^/L)	CON	0.9 ± 0.1	1.3 ± 0.1 ^Aa^	1.0 ± 0.1 ^Aa^	0.859	0.698	0.493
	BS_50_	1.0 ± 0.1	1.3 ± 0.1 ^Aa^	1.2 ± 0.1 ^Aa^			
	BS_100_	0.8 ± 0.1	1.0 ± 0.1 ^Aa^	1.0 ± 0.1 ^Aa^			
Lymphocytes (10^9^/L)	CON	6.6 ± 0.2	8.2 ± 0.4 ^Aa^	6.9 ± 0.2 ^Ba^	0.103	0.692	0.168
	BS_50_	6.9 ± 0.3	8.3 ± 0.5 ^Aa^	6.0 ± 0.5 ^Ba^			
	BS_100_	5.9 ± 0.6	4.9 ± 0.2 ^Aa^	4.8 ± 0.3 ^Aa^			

^1^ CON—control group of cows; BS_50_—group of cows supplemented with 50 mL of brown seaweed (10% *A. nodosum*, 10%); BS_100_—group of cows supplemented with 100 mL of brown seaweed (10% *A. nodosum*); ^2^ P1—the one-week period before brown seaweed supplementation; P2—the one-week period after 15 days of brown seaweed supplementation; P3—the one week after a total of 30 days of brown seaweed supplementation; ^3^ significance was declared at *p* < 0.05; T—treatment; P—period; T × P—interaction between treatment and period; RBCs—red blood cells; MCV—mean corpuscular volume; MCH—mean corpuscular hemoglobin; MCHC—mean corpuscular hemoglobin concentration; WBC—total white blood cell count; ^AB^ different uppercase letters indicate statistically significant differences (*p* < 0.05) within the same group at different periods; ^ab^ different lowercase letters indicate statistically significant differences (*p* < 0.05) between groups at the same period.

**Table 4 animals-14-01520-t004:** Mean ± SE and ANOVA *p*-values for the effects of brown seaweed (10% *Ascophyllum nodosum*) supplementation on blood biochemical parameters.

Parameter (Unit)	Group ^1^	Period of the Experiment ^2^			*p*-Value ^3^		
		P1	P2	P3	T	P	T × P
Total protein (g/L)	CON	86.8 ± 4.2	94.1 ± 4.5 ^Aa^	92.4 ± 4.3 ^Aa^	0.145	0.716	0.170
	BS_50_	88.5 ± 1.3	92.2 ± 1.2 ^Aa^	89.1 ± 1.8 ^Aa^			
	BS_100_	88.8 ± 2.1	90.5 ± 2.3 ^Aa^	92.8 ± 2.4 ^Aa^			
Albumin (g/L)	CON	42.4 ± 1.7	44.4 ± 1.9 ^Aa^	43.0 ± 1.9 ^Aa^	0.733	0.757	0.115
	BS_50_	46.3 ± 1.3	47.3 ± 2.1 ^Aa^	44.6 ± 1.5 ^Aa^			
	BS_100_	41.2 ± 1.8	40.5 ± 1.9 ^Aa^	41.8 ± 2.0 ^Aa^			
BUN (mmol/L)	CON	8.66 ± 0.3	8.10 ± 0.5 ^Aa^	7.36 ± 0.6 ^Aa^	0.738	0.232	0.347
	BS_50_	8.66 ± 0.4	8.46 ± 0.3 ^Aa^	7.08 ± 0.2 ^Aa^			
	BS_100_	9.72 ± 0.8	7.57 ± 0.4 ^Aa^	7.68 ± 0.5 ^Aa^			
Total bilirubin (µmol/L)	CON	1.65 ± 0.1	1.21 ± 0.1 ^Aa^	1.89 ± 0.2 ^Aa^	0.060	0.405	0.312
	BS_50_	2.17 ± 0.2	1.91 ± 0.2 ^Ab^	2.54 ± 0.3 ^Aa^			
	BS_100_	1.66 ± 0.2	0.48 ± 0.1 ^Ac^	1.82 ± 0.2 ^Ba^			
TAG (mmol/L)	CON	0.15 ± 0.02	0.13 ± 0.02 ^Aa^	0.21 ± 0.02 ^Aa^	0.079	0.442	0.018
	BS_50_	0.21 ± 0.02	0.21 ± 0.02 ^Aa^	0.13 ± 0.01 ^Bb^			
	BS_100_	0.14 ± 0.03	0.12 ± 0.01 ^Aa^	0.15 ± 0.03 ^Aab^			
Total cholesterol (mmol/L)	CON	5.79 ± 0.6	6.70 ± 0.5 ^Aa^	6.71 ± 0.6 ^Aa^	0.630	0.707	0.749
	BS_50_	6.84 ± 0.6	7.81 ± 0.6 ^Aa^	7.61 ± 0.6 ^Aa^			
	BS_100_	5.94 ± 0.5	6.55 ± 0.6 ^Aa^	6.35 ± 0.5 ^Aa^			
HDL-C (mmol/L)	CON	149.7 ± 12.6	166.9 ± 7.2 ^Aa^	191.1 ± 11.3 ^Ba^	0.310	0.380	0.021
	BS_50_	154.3 ± 6.7	195.0 ± 9.2 ^Ab^	190.1 ± 12.2 ^Aa^			
	BS_100_	135.9 ± 7.4	140.3 ± 8.9 ^Aa^	181.1 ± 14.1 ^Ba^			
Glucose (mmol/L)	CON	3.72 ± 0.1	3.02 ± 0.2 ^Aa^	3.13 ± 0.2 ^Aa^	0.643	0.806	0.394
	BS_50_	3.58 ± 0.1	2.98 ± 0.2 ^Aa^	3.55 ± 0.2 ^Aa^			
	BS_100_	3.58 ± 0.1	3.02 ± 0.2 ^Aa^	3.33 ± 0.1 ^Aa^			
BHB (mmol/L)	CON	0.41 ± 0.02	0.66 ± 0.04 ^Aa^	0.63 ± 0.05 ^Aa^	0.153	0.464	0.216
	BS_50_	0.37 ± 0.03	0.86 ± 0.11 ^Aa^	0.65 ± 0.06 ^Ba^			
	BS_100_	0.44 ± 0.04	0.81 ± 0.06 ^Aa^	0.68 ± 0.06 ^Aa^			
NEFA (mmol/L)	CON	0.46 ± 0.02	0.39 ± 0.12 ^Aa^	0.29 ± 0.03 ^Aa^	0.191	0.376	0.130
	BS_50_	0.53 ± 0.03	0.37 ± 0.02 ^Aa^	0.36 ± 0.06 ^Aa^			
	BS_100_	0.46 ± 0.06	0.19 ± 0.03 ^Aa^	0.32 ± 0.02 ^Aa^			
γ-GT (U/L)	CON	33.5 ± 2.7	33.9 ± 4.9 ^Aa^	32.2 ± 4.7 ^Aa^	0.994	0.714	0.621
	BS_50_	47.9 ± 4.6	44.9 ± 8.3 ^Aa^	43.0 ± 6.8 ^Aa^			
	BS_100_	45.9 ± 5.2	41.9 ± 4.7 ^Aa^	42.7 ± 5.9 ^Aa^			
AST (U/L)	CON	127.9 ± 14.8	112.4 ± 9.2 ^Aa^	105.7 ± 4.1 ^Aa^	0.890	0.254	0.312
	BS_50_	148.6 ± 12.8	125.2 ± 5.9 ^Aa^	114.6 ± 9.5 ^Aa^			
	BS_100_	163.0 ± 10.4	119.0 ± 9.5 ^Aa^	128.5 ± 11.6 ^Aa^			

^1^ CON—control group of cows; BS_50_—group of cows supplemented with 50 mL of brown seaweed (10% *A. nodosum*, 10%); BS_100_—group of cows supplemented with 100 mL of brown seaweed (10% *A. nodosum*); ^2^ P1—the one-week period before brown seaweed supplementation; P2—the one-week period after 15 days of brown seaweed supplementation; P3—the one-week period after a total of 30 days of brown seaweed supplementation; ^3^ significance was declared at *p* < 0.05; T—treatment; P—period; T × P—interaction between treatment and period; BUN—blood urea nitrogen; TAG—triacylglycerols; HDL-C—HDL-cholesterol; BHB—beta-hydroxybutyrate; NEFAs—non-esterified fatty acids; γ-GT—gamma-glutamyltransferase; AST—aspartate aminotransferase; ^AB^ different uppercase letters indicate statistically significant differences (*p* < 0.05) within the same group at different periods; ^ab^ different lowercase letters indicate statistically significant differences (*p* < 0.05) between groups at the same period.

**Table 5 animals-14-01520-t005:** Mean ± SE and ANOVA *p*-values for the effects of brown seaweed supplementation (10% *Ascophyllum nodosum*) on the chemical composition of milk, milk fatty acid profile, and somatic cell count.

Parameter	Group ^1^	Period of the Experiment ^2^			*p*-Value ^3^		
		P1	P2	P3	T	P	T × P
Chemical composition of milk (%)							
Protein	CON	3.37 ± 0.1	3.02 ± 0.1 ^Aa^	3.21 ± 0.1 ^Ba^	0.607	0.451	0.642
	BS_50_	3.52 ± 0.1	3.30 ± 0.1 ^Aa^	3.38 ± 0.1 ^Aa^			
	BS_100_	3.50 ± 0.1	3.16 ± 0.1 ^Aa^	3.28 ± 0.1 ^Aa^			
Fat	CON	3.40 ± 0.1	3.41 ± 0.1 ^Aa^	3.28 ± 0.2 ^Aa^	0.315	0.207	0.120
	BS_50_	3.43 ± 0.1	3.42 ± 0.1 ^Aa^	3.60 ± 0.1 ^Ab^			
	BS_100_	3.41 ± 0.1	3.35 ± 0.1 ^Aa^	3.50 ± 0.1 ^Aab^			
Lactose	CON	4.92 ± 0.1	4.42 ± 0.1 ^Aa^	4.69 ± 0.1 ^Aa^	0.361	0.039	0.432
	BS_50_	5.17 ± 0.1	4.49 ± 0.2 ^Aa^	4.57 ± 0.1 ^Aa^			
	BS_100_	5.22 ± 0.2	4.42 ± 0.3 ^Aa^	4.74 ± 0.1 ^Aa^			
Total solids	CON	11.69 ± 0.1	10.85 ± 0.3 ^Aa^	11.18 ± 0.2 ^Aa^	0.541	0.943	0.867
	BS_50_	12.12 ± 0.1	11.21 ± 0.3 ^Aa^	11.54 ± 0.1 ^Aa^			
	BS_100_	12.18 ± 0.2	10.92 ± 0.4 ^Aa^	11.52 ± 0.2 ^Aa^			
SNF	CON	8.29 ± 0.1	7.44 ± 0.2 ^Aa^	7.90 ± 0.2 ^Aa^	0.513	0.517	0.593
	BS_50_	8.69 ± 0.1	7.79 ± 0.3 ^Aa^	7.94 ± 0.1 ^Aa^			
	BS_100_	8.77 ± 0.2	7.57 ± 0.4 ^Aa^	8.02 ± 0.1 ^Aa^			
Milk fatty acid profile (% of fatty acids)							
C4:0	CON	0.88 ± 0.11	1.74 ± 0.17 ^Aa^	1.07 ± 0.16 ^Aa^	0.085	0.884	0.144
	BS_50_	0.77 ± 0.09	1.13 ± 0.09 ^Ab^	1.05 ± 0.10 ^Aa^			
	BS_100_	1.03 ± 0.09	1.73 ± 0.09 ^Aa^	1.21 ± 0.20 ^Aa^			
C6:0	CON	1.27 ± 0.16	2.39 ± 0.27 ^Aa^	1.57 ± 0.23 ^Ba^	0.252	0.987	0.056
	BS_50_	1.10 ± 0.12	1.52 ± 0.09 ^Ab^	1.54 ± 0.09 ^Aa^			
	BS_100_	1.34 ± 0.10	2.15 ± 0.12 ^Aab^	1.64 ± 0.24 ^Aa^			
C8:0	CON	1.15 ± 0.14	2.18 ± 0.27 ^Aa^	1.42 ± 0.21 ^Aa^	0.236	0.516	0.021
	BS_50_	1.04 ± 0.09	1.40 ± 0.10 ^Ab^	1.38 ± 0.07 ^Aa^			
	BS_100_	1.15 ± 0.11	1.87 ± 0.16 ^Aab^	1.45 ± 0.19 ^Aa^			
C10:0	CON	2.99 ± 0.29	5.00 ± 0.65 ^Aa^	3.65 ± 0.47 ^Ba^	0.450	0.699	0.039
	BS_50_	2.79 ± 0.22	3.68 ± 0.32 ^Aa^	3.65 ± 0.23 ^Aa^			
	BS_100_	3.24 ± 0.39	4.38 ± 0.44 ^Aa^	3.57 ± 0.54 ^Aa^			
C11:0	CON	0.21 ± 0.03	0.44 ± 0.06 ^Aa^	0.38 ± 0.05 ^Aa^	0.111	0.461	0.278
	BS_50_	0.20 ± 0.02	0.32 ± 0.03 ^Aa^	0.35 ± 0.03 ^Aa^			
	BS_100_	0.24 ± 0.03	0.42 ± 0.05 ^Aa^	0.31 ± 0.05 ^Aa^			
C12:0	CON	3.19 ± 0.18	4.18 ± 0.43 ^Aa^	3.55 ± 0.34 ^Ba^	0.453	0.929	0.010
	BS_50_	3.15 ± 0.22	3.58 ± 0.25 ^Aa^	3.98 ± 0.30 ^Aa^			
	BS_100_	3.56 ± 0.40	3.95 ± 0.41 ^Aa^	3.46 ± 0.43 ^Aa^			
C14:0	CON	9.60 ± 0.30	10.30 ± 0.62 ^Aa^	10.2 ± 0.57 ^Aa^	0.474	0.716	0.114
	BS_50_	9.90 ± 0.41	9.95 ± 0.29 ^Aa^	11.2 ± 0.48 ^Ba^			
	BS_100_	10.90 ± 0.74	10.70 ± 0.74 ^Aa^	10.6 ± 0.91 ^Aa^			
C14:1	CON	0.65 ± 0.07	0.58 ± 0.08 ^Aa^	0.82 ± 0.11 ^Aa^	0.295	0.818	0.641
	BS_50_	0.76 ± 0.17	0.76 ± 0.07 ^Aa^	0.87 ± 0.13 ^Aa^			
	BS_100_	0.72 ± 0.09	0.75 ± 0.09 ^Aa^	0.97 ± 0.19 ^Aa^			
C15:0	CON	1.13 ± 0.09	0.85 ± 0.07 ^Aa^	0.97 ± 0.09 ^Aa^	0.773	0.408	0.981
	BS_50_	1.09 ± 0.10	0.85 ± 0.06 ^Aa^	1.02 ± 0.13 ^Aa^			
	BS_100_	1.40 ± 0.12	0.91 ± 0.1 ^Aa^	1.01 ± 0.06 ^Aa^			
C16:0	CON	31.3 ± 0.75	30.9 ± 0.59 ^Aa^	31.8 ± 0.52 ^Aa^	0.023	0.086	0.215
	BS_50_	32.2 ± 0.32	31.9 ± 0.47 ^Aa^	35.0 ± 0.44 ^Bb^			
	BS_100_	33.0 ± 0.71	31.8 ± 0.60 ^Aa^	33.5 ± 1.53 ^Bb^			
C16:1	CON	1.17 ± 0.34	0.91 ± 0.14 ^Aa^	0.84 ± 0.08 ^Aa^	0.171	0.052	0.031
	BS_50_	1.48 ± 0.23	0.89 ± 0.09 ^Aa^	1.32 ± 0.11 ^Bb^			
	BS_100_	1.14 ± 0.19	0.96 ± 0.12 ^Aa^	1.02 ± 0.17 ^Aab^			
C17:0	CON	0.58 ± 0.01	0.26 ± 0.05 ^Aa^	0.40 ± 0.05 ^Aa^	0.057	0.466	0.096
	BS_50_	0.34 ± 0.02	0.34 ± 0.03 ^Aa^	0.33 ± 0.04 ^Aab^			
	BS_100_	0.37 ± 0.04	0.29 ± 0.05 ^Aa^	0.24 ± 0.01 ^Ab^			
C17:1	CON	0.55 ± 0.08	0.26 ± 0.06 ^Aa^	0.43 ± 0.07 ^Ba^	0.412	0.064	0.129
	BS_50_	0.42 ± 0.08	0.50 ± 0.05 ^Ab^	0.32 ± 0.04 ^Ba^			
	BS_100_	0.58 ± 0.07	0.28 ± 0.04 ^Aab^	0.40 ± 0.09 ^Aa^			
C18:0	CON	10.3 ± 0.98	9.32 ± 0.40 ^Aa^	10.9 ± 1.00 ^Aa^	0.576	0.867	0.060
	BS_50_	9.96 ± 1.03	10.6 ± 0.77 ^Aa^	7.70 ± 0.63 ^Bb^			
	BS_100_	8.77 ± 0.93	9.09 ± 0.97 ^Aa^	8.37 ± 0.39 ^Ab^			
C18:1 n-9c	CON	25.6 ± 0.84	21.2 ± 1.17 ^Aa^	22.0 ± 0.97 ^Aa^	0.590	0.858	0.265
	BS_50_	26.8 ± 1.03	22.3 ± 0.99 ^Aa^	21.4 ± 0.91 ^Aa^			
	BS_100_	25.6 ± 1.93	19.9 ± 1.31 ^Aa^	20.9 ± 2.36 ^Aa^			
C18:1 n-11t	CON	1.45 ± 0.27	1.25 ± 0.25 ^Aa^	1.44 ± 0.37 ^Aa^	0.333	0.193	0.088
	BS_50_	1.75 ± 0.18	1.48 ± 0.23 ^Aa^	0.71 ± 0.12 ^Aa^			
	BS_100_	1.32 ± 0.20	1.24 ± 0.22 ^Aa^	1.33 ± 0.25 ^Aa^			
C18:2 n-6	CON	5.17 ± 0.34	4.78 ± 0.19 ^Aa^	4.85 ± 0.07 ^Aa^	0.215	0.072	0.880
	BS_50_	5.34 ± 0.18	4.96 ± 0.19 ^Aa^	5.27 ± 0.13 ^Ab^			
	BS_100_	4.97 ± 0.25	4.63 ± 0.30 ^Aa^	4.46 ± 0.19 ^Aa^			
C18:3 n-3	CON	0.58 ± 0.13	0.44 ± 0.03 ^Aa^	0.50 ± 0.07 ^Aa^	0.007	0.795	0.364
	BS_50_	0.44 ± 0.02	0.65 ± 0.13 ^Aa^	0.38 ± 0.04 ^Aa^			
	BS_100_	0.50 ± 0.04	0.83 ± 0.22 ^Aa^	0.76 ± 0.09 ^Ab^			
C18:2 (CLA)	CON	0.46 ± 0.28	0.42 ± 0.04 ^Aa^	0.42 ± 0.08 ^Aa^	0.179	0.490	0.421
	BS_50_	0.34 ± 0.06	0.63 ± 0.08 ^Aa^	0.51 ± 0.05 ^Aa^			
	BS_100_	0.21 ± 0.04	0.52 ± 0.14 ^Aa^	0.76 ± 0.25 ^Aa^			
SFA	CON	62.7 ± 1.71	67.4 ± 1.52 ^Aa^	65.9 ± 0.85 ^Aa^	0.618	0.269	0.098
	BS_50_	62.5 ± 1.26	65.3 ± 1.03 ^Aa^	67.1 ± 0.90 ^Aa^			
	BS_100_	65.0 ± 1.62	67.3 ± 1.53 ^Aa^	65.4 ± 2.39 ^Aa^			
MUFA	CON	29.4 ± 1.25	24.0 ± 1.48 ^Aa^	25.3 ± 0.70 ^Aa^	0.939	0.246	0.312
	BS_50_	30.6 ± 0.94	25.9 ± 1.07 ^Aa^	24.7 ± 0.78 ^Aa^			
	BS_100_	28.7 ± 1.53	23.2 ± 1.29 ^Aa^	24.7 ± 2.33 ^Aa^			
PUFA	CON	6.22 ± 0.36	5.64 ± 0.21 ^Aa^	5.77 ± 0.15 ^Aa^	0.122	0.063	0.618
	BS_50_	6.12 ± 0.23	6.24 ± 0.26 ^Aa^	6.15 ± 0.14 ^Aa^			
	BS_100_	5.68 ± 0.25	5.99 ± 0.38 ^Aa^	6.00 ± 0.32 ^Aa^			
Milk somatic cell count (cell/mL)							
Log_10_ SCC	CON	5.29 ± 0.17	5.49 ± 0.21 ^Aa^	5.47 ± 0.20 ^Aa^	0.248	0.936	0.807
	BS_50_	5.02 ± 0.25	5.08 ± 0.19 ^Aa^	5.16 ± 0.29 ^Aa^			
	BS_100_	5.13 ± 0.19	5.20 ± 0.31 ^Aa^	4.93 ± 0.14 ^Aa^			

^1^ CON—control group of cows; BS_50_—group of cows supplemented with 50 mL of brown seaweed (10% *A. nodosum*, 10%); BS_100_—group of cows supplemented with 100 mL of brown seaweed (10% *A. nodosum*); ^2^ P1—the one-week period before brown seaweed supplementation; P2—the one-week period after 15 days of brown seaweed supplementation; P3—the one-week period after total of 30 days of brown seaweed supplementation; ^3^ significance was declared at *p* < 0.05; T—treatment; P—period; T × P—interaction between treatment and period; SNF—solid non-fat; CLA—conjugated linoleic acid; SFAs—saturated fatty acids; MUFAs—monounsaturated fatty acids; PUFAs—polyunsaturated fatty acids; SCC—somatic cell count; ^AB^ different uppercase letters indicate statistically significant differences (*p* < 0.05) within the same group at different periods; ^ab^ different lowercase letters indicate statistically significant differences (*p* < 0.05) between groups at the same period.

## Data Availability

The data presented in this study are available upon request from the corresponding author and are available at https://mitimetcattle.vet.bg.ac.rs/ (accessed on 25 March 2024).

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
