# Peer review of "Effects of Brown Seaweed (*Ascophyllum nodosum*) Supplementation on Enteric Methane Emissions, Metabolic Status and Milk Composition in Peak-Lactating Holstein Cows"

_animals, 2024, doi:10.3390/ani14111520_

Round 1
Reviewer 1 Report
Comments and Suggestions for Authors
The manuscript provides some interesting information on the effect of A. nodosum on enteric methane emissions, metabolic status and milk composition in lactating cows.
However, I must raise some question especially about the methods used in the presented study.
Abstract:
L41-42: I think this statement should be formulated more carefully as the effect on DMI was numerically low and only for one group in P2 and did not last over the whole trial.
L108: How were the groups separated throughout the trial?
L113: How did you ensure proper mixing of the diets and and that there is no carryover of the substance when mixing the different rations?
L.128-123: Belongs to the results section.
Table 1: Please give the diet information in % of DM instead of kg of DM/day.
L151: Was the feed presented in troughs and individual to each animal or on the floor and measured for every group? How was it ensured that only the animals in the relevant group had access to the feed? Please specify feeding situation.
Results: the results should be described more clearly in the text for each measured value. The general statement that a measured value was statistically influenced by this or that effect, without saying in which direction and to what extent, has little informative value.
L266-L274: See my general comment for the results section above. Paragraph can be omitted.
L433-435: I think you should also discuss the pros and cons of the technique that was used in the presented experiment. Is there any calibration of the instrument used to measure methane concentration in your study?
L577-579: Should be formulated more carefully. Results on that are not so clear in this case.
Author Response
Thank you very much for taking the time to review this manuscript. Please find the detailed responses in the attached document and the corresponding corrections highlighted in the re-submitted files. We have considered all comments in detail and have tried to soften, correct and align any contentious points to the requirements of your expert opinion in the revised version of the manuscript.

Reviewer 2 Report
Comments and Suggestions for Authors
The study has a lofty aim and clear objectives. Well designed and parameters measured are very relevant to the study's aims. However, methodological concerns arise from the methane measurement device. Statistical analysis and interpretation need to be corrected. Specific comments are detailed below:
How did the investigators ensure that the total volume of seaweed supplement was consumed daily considering it was added to the TMR feed on offer and fed ad lib?
L167-173: what is the reliability of this methane measurement protocol. Considering cows will wiggle their heads around, thus altering the position of the nostrils, even if the detector device is fixed. This also does not account for ambient methane which can be higher or lower depending on how many cows is around at that point. Will all the confounding sources of errors be small enough as to not affect values and comparisons?
L174-177: how does this background value come about? Methane in air at any particular point will be what is diffused from nearby animals.
For most result, there is significance with interaction, treatment and period effect. The descriptions in L272-283 needs to be improved.
For example, P1 is background data. Repeating the values for CON, BS50 and BS100 confounds the analysis. Looking at P2 and P3, it is clear that “methane production (ppm) was significantly reduced by BS50 and BS100 and greater reduction was achieved at P3 than P2”. That clearly describes a pattern observable based on the two factors. Authors have only told us there exist some effect but have not described this effect.
L298-301: “influenced by period and affected by dietary treatment..”. This has not communicated the data clearly enough. See the previous comment.
L306-312: see previous comment.
For table 3 and 4, Data obtained in P1 should form a background data and the average should be reported regardless of treatment.
A 2 (P2, P3) X 3 (CON, BS50, BS100) factorial plus control (average of P1 data regardless of treatments) analysis method may be more appropriate for this type of study. I suggest authors discuss with a statistician.
L348-356: Where there is significant interactions, the interaction effect is discussed at the expense of the simple main effects of the factors. Only when interaction is not significant can the attention shift towards the simple main effect of period or treatment where either is significant.
L427-436: Authors have simply tried to downplay the result of Ataya et al, which uses GreenFeed systems in methane monitoring compared to the device used by the current authors. However, the argument falls flat on logic. The concentration of a gas is a function of the volume of the gas in a volume of air. With green feed, animals enter a box (voluntarily) whose dimensions are known. Air concentration stabilises and air is sampled and concentration determined. Based on the validity of the greenfeed system, methane measurement conducted by authors is very doubtful in terms of accuracy and repeatability.
L443-446: what is the particle size of the current product? Furthermore, with a TMR ration offered ad lib it is expected that continuous release of active ingredients into the rumen environment occurs thus particle size will be a minor issue. Authors fed the seaweed as component of the morning TMR.
L448-449: Authors also should factor that longer duration means microbial adaptation. Whereas period P2 and the referenced study are somewhat similar.
In my opinion, the method of analysis- methane sampling as well as statistical analysis has exaggerated the outcomes of this study and this warrants re-analysis (statistics).
L509-515: Milk fat and methane are closely correlated parameters. Authors have reported increased milk fat but also significant reduction in methane. BY what mechanism, it has not be clarififed.
Author Response

(The authors gave the same response as above.)

Round 2
Reviewer 2 Report
Comments and Suggestions for Authors
The authors have taken into account the comments from previous peer review, particularly the statistical prcedure, analysis and interpretation of the data.
The outcome has also resulted in modifications in the result and discussion session which is satisfactory and will make for a good addition to knowledge.